# Programmed Cell Death in Cystinosis

**DOI:** 10.3390/cells11040670

**Published:** 2022-02-15

**Authors:** Elizabeth G. Ames, Jess G. Thoene

**Affiliations:** Division of Pediatric Genetics, Metabolism, and Genomic Medicine, Department of Pediatrics, University of Michigan Health System, Ann Arbor, MI 48109, USA; amese@med.umich.edu

**Keywords:** cystinosis, apoptosis, programmed cell death

## Abstract

Cystinosis is a lethal autosomal recessive disease that has been known clinically for over 100 years. There are now specific treatments including dialysis, renal transplantation and the orphan drug, cysteamine, which greatly improve the duration and quality of patient life, however, the cellular mechanisms responsible for the phenotype are unknown. One cause, programmed cell death, is clearly involved. Study of extant literature via Pubmed on “programmed cell death” and “apoptosis” forms the basis of this review. Most of such studies involved apoptosis. Numerous model systems and affected tissues in cystinosis have shown an increased rate of apoptosis that can be partially reversed with cysteamine. Proposed mechanisms have included changes in protein signaling pathways, autophagy, gene expression programs, and oxidative stress.

## 1. Introduction

Programmed cell death is such an essential cell function that it is uniquely challenging to dissect out the contributions to the development of the complex clinical phenotype of cystinosis. Here we present a summary and analysis of literature pertaining to the role of apoptosis in cystinosis obtained by searching PubMed for all articles identified with the searches “cystinosis and apoptosis” as well as “cystinosis and programmed cell death”.

Cystinosis is a pan-systemic disease which causes severe failure to thrive, retinopathy, keratopathy, renal Fanconi syndrome, and progressive renal dysfunction that results in renal failure by age 10 years. It was first described in 1903 in two sibs as “Familiare Cystindiathese”. In addition to severe failure to thrive, these children displayed increased urination and ultimately kidney failure. On microscopic examination at autopsy, the children were found to have prominent tissue crystals, identified as cystic oxide, now known to be cystine [1]. In 1967 it was demonstrated that the cystine was intracellular within leukocytes from patients with cystinosis, and that the cystine content reflected a Mendelian distribution [2], which enabled discovery of defective cystine lysosomal transport as the genesis of the disease in 1982 [3].

This was followed by development of cysteamine as the first effective specific therapy for cystinosis [4]. It is effective because cysteamine reacts with cystine within the lysosomes to form a product exported by the intact lysine carrier [5], reducing the cystine concentration to the level of the obligate heterozygous parents who do not develop any of the phenotypic features. That transporter has been characterized as closely related to the yeast PQ loop family of vacuolar membrane transporters, and designated PQLC2 [6]

Cysteamine, one of the first US FDA-approved orphan drugs (1994), has a beneficial effect on the course of the disease (see elsewhere in this paper). However, as noted by many authors, cysteamine does not prevent renal failure, but merely delays it, nor does it improve the renal Fanconi Syndrome [7].

## 2. The Role of Apoptosis in Cystinosis and Response to Cysteamine

Failure of cystinosin results in cystine accumulation that is confined to lysosomes and precludes cystine from interacting with intracellular metabolism. Therefore, the first obvious challenge is to determine how cystine could interfere to such an extent as to cause the lethal phenotype. A first step toward explaining this enigma came in 2002 [8] with the observation that cystinotic fibroblasts display a 2 to 3-fold increase in the apoptotic rate compared to normal fibroblasts, and that normalization of the cystine content by pre-treatment with cysteamine also normalizes the apoptosis rate. Moreover, when normal cells are pre-loaded with cystine dimethylester to increase the lysosomal cystine content to the levels seen in cystinotic cells, the apoptosis rate increases to that seen in cystinotic cells. Similar results were found in renal proximal tubule epithelial cells as well [8]. The toxicity of CDME in normal cultured fibroblasts via a toxic effect on mitochondria has been raised [9]. However this study employed 1–5 mM CDME, whereas the study cited above showing increased apoptosis after loading normal cells with CDME used 0.25–0.5 mmM CDME. This lower concentration showed little effect on cells in the Wilmer study.

The next step occurred in 2006 when it was shown that due to lysosomal permeabilization, which occurs early in the apoptosis cascade, and before the cell is irreversibly committed to cell death, cystine can leave the lysosomes and thiolate a crucial disulfide in the proapoptotic kinase, PKC∂. It had previously been shown that such cysteinylation increases the activity of this kinase [10]. Further, inhibition of PKC∂ via siRNA silencing or with 12-O-tetradecanoylphorbol-13-acetate treatment markedly diminishes the apoptotic rate in parallel with the diminution in kinase activity [11]. A plot of lysosomal cystine concentration versus the rate of apoptosis including both normal and cystinotic cultured fibroblasts has the form of a rectangular hyperbola with a K_m_ of 0.2 nmol/mg protein, which approximates the normal lysosomal cystine content. Fibroblast lines from intermediate and ocular cystinosis do not lie on this curve, but display less apoptosis for a given cystine content [12].

Currently, increased apoptosis in cystinotic tissues has been described by several laboratories, which have confirmed that decreasing lysosomal cystine storage also decreases the elevated rate of apoptosis [7,8,13]. One group found a relatively minor increase in apoptosis (1.4-fold) in stem cells and organoids which was not reduced by cysteamine treatment [14]. This raises the issue of how well model systems mimic the clinical, biochemical, and cellular phenotypes, and which of these cell and animal systems are most faithful to the human disease. Apoptosis studies have been conducted in genetically-created cystinotic systems or via utilization of cystine dimethylester to create lysosomal cystine loading in human, mouse, and rat fibroblasts [12] (See Table). A study showing an increased rate of apoptosis in cystinotic proximal tubule cells found glutathione (GSH) depletion and failure to increase mitochondrial activity and maintain ATP levels under hypoxic conditions [13]. Increased apoptosis with correction by cysteamine was shown in the zebrafish model of cystinosis [7]. Using this model, this study showed elevated lysosomal cystine, increased apoptosis, and tubular dysfunction progressing to renal failure. Similar findings were reported in human induced pluripotent stem cells and kidney organoids [14] in which significantly elevated cystine and increased apoptosis was observed. Cystine was decreased by treatment with cysteamine, but the elevated rate of apoptosis was not. The cells displayed enlarged lysosomes, attributed to the osmotic effect of cystine, because treatment with cysteamine partially restored lysosomal size. However, since cystine is the least soluble amino acid (112 mg/L at 25 degrees centigrade and neutral pH, and has minimum solubility at the lysosomal pH of 5.0 [15], it precipitates at 0.46 mM. Induced vacuolation of lysosomes using membrane-impermeant sucrose requires 24 h exposure at 100 mM [16], hence the enlarged lysosomes may have occurred from other effects. No abnormalities in ATP or GSH concentration were found in these cells.

Increased apoptosis in cystinotic tissues can account for much of the clinical phenotype seen in cystinosis. Progressive loss of renal proximal tubular epithelial cells via abnormal apoptosis results in the “Swan Neck” deformity [17], and such progression can subsequently lead to non-functional atubular glomeruli [18] that can progress to overt renal failure over time. The renal tubule and retina are highly sensitive to apoptosis [8], and these two tissues are the first to be affected in nephropathic cystinosis [1], hence it is feasible that the order of tissues involved in the disease reflects the intrinsic sensitivity of each to apoptosis. It should be noted that abnormal apoptosis contributes to many birth defects [19], however dysmorphism is not part of the cystinosis phenotype. It therefore appears that increased apoptosis in cystinotic tissues is a regulated multi-step process, not just an “ON” switch for immediate cell death.

## 3. Autophagy in Cystinosis

Autophagy is a specialized cellular process that can both promote survival of a cell or promote programmed cell death. Autophagy has an entwined relationship with apoptosis due to shared regulatory genes and cellular machinery, which can confound experimental isolation of one cell death mechanism compared to another [20]. The role of autophagy in cystinosis has been specifically reviewed elsewhere, but there are several studies that examine the interplay between apoptosis and autophagy within model systems of cystinosis. One of the most basic observations initially showed that autophagy was increased in cystinotic tissues. Renal biopsies and cultured fibroblasts identified morphological and protein signaling markers of increased mitochondrial autophagy and autophagosomes while observing fewer total mitochondria in patients with nephropathic cystinosis compared to other forms of cystinosis and normal cells. The use of an autophagy inhibitor (3-methyl adenine) normalized the increased rate of apoptosis seen in the cystinotic cell lines. These results are consistent with autophagy both leading to and promoting apoptosis in cystinosis [21].

In a screen for compounds which reduce levels of specific markers of impaired autophagy (p62/SQSTM1), luteolin was shown to ameliorate several of the phenotypic abnormalities in constitutively-induced *CTNS*-null renal proximal tubular epithelial cells, including the aberrant autophagy-lysosomal degradative pathway, abnormalities in redox status, and increased sensitivity to apoptosis [22].

Analysis of renal biopsies and cultured fibroblasts identified morphological and protein signaling markers of mitochondrial autophagy in patients with nephropathic cystinosis compared to other forms of cystinosis and normal cells. These findings led to the hypothesis that autophagy leads to and synergistically promotes apoptosis in cystinosis [21].

## 4. Gene Expression Profiling and Signal Cascades

Gene expression studies of blood samples from cystinosis patients shows increased gene expression of apoptosis, mitochondrial dysfunction, and oxidative stress markers [23].

Rossi et al. reported that *NLRP2* is overexpressed in cystinotic proximal tubule epithelial cells compared to healthy subject cells. Using *NLRP2* overexpression, they noted a lower apoptotic rate. When *NLRP2* was silenced with siRNA, the apoptotic rate increased, which seems to contradict what is commonly observed in cystinosis [24].

## 5. Oxidative Stress

The issue of abnormal redox potential in cystinotic cells and tissues, and more recently, as a mediator of aberrant apoptosis, is challenging due to the multiple systems employed and interpretation of results. In 1978, Oshima et al. reported the GSH content of cystinotic fibroblasts was 91% of the content found in normal fibroblasts [25]. In three cystinotic fibroblast lines, Chol et al. found GSH content was 70% of that found in control cells [26]. Studies in matched, low passage number cystinotic fetal skin, cystinotic fetal lung, and normal fetal lung fibroblasts, found the concentration of GSH, and the GSH/GSSG ratio were comparable [27]. It is not expected that the abnormally increased cystine accumulations in cystinotic cells would alter cellular redox potential given that the lysosomal cystine pool is physically separated from the cystosolic compartment. Without functioning cystinosin, cystine remains intralysosomal until exocystosed. Lysosomes represent only about 10% of cell volume in cystinotic fibroblasts [28], and since cystine is very insoluble, with a solubility limit of about 0.5 mM in aqueous solution to 1.66 mM in plasma [1], instantaneous dilution into the total cytosol of a saturating solution of cystine would yield a final cytosolic cystine concentration of 0.05–0.166 mM. Given the GSH concentration in the cytosol of ~10 mM [29], it is unlikely that cystine would greatly alter the global redox status of the cytosol. Any GSSG formed by reaction with cystine would be rapidly reduced back to GSH by cytosolic reductases [1]. These observations were extended by Wilmer et al. who found, using a proximal tubular epithelial cell model, no difference in GSH between control and cystinotic cells, ATP production, or the oxidation state of proteins or lipids. They also found that cysteamine increased GSH in both normal and control cells, but did not improve sodium-dependent phoshate uptake. This last finding is consistent with the clinical observation that cysteamine therapy does not improve the renal Fanconi syndrome, a component of which is failure of sodium reabsorption [30]. Bellomo et al. showed that in HK-2 cells CTNS gene expression was correlated with intracellular cysteine level and with the cysteine/cystine equilibrium, implying a regulated role for CTNS in maintaining the intracellular redox state [31].

Studies by Sumayao et al. involving redox potential and apoptosis used a variety of cell and tissue models. With *CTNS* silencing of a proximal tubular epithelial cell line, they found an increase in apoptosis and ROS which was decreased by cysteamine [32]. GSH was reported in an earlier publication [33] as being two-fold decreased in the *CTNS* knockout cells. However, GSH was measured using a non-specific reagent, 5,5′-dithio-bis-2-nitrobenzoic acid, which reacts with all thiols. It is also not surprising that the concentration of ROS is diminished in the presence of cysteamine, since it is a known free-radical scavenger that is oxidized to the disulfide, cystamine, which can then be regenerated to cysteamine by GSH [34].

A recent paper shows that a disulfide, disulfiram, forms mixed disulfides with cystine, and has cystine-depleting and anti-apoptotic effects [35] Presumably, like the disulfide cystamine [4], reduction of disulfiram to the free thiol diethyldithiocarbamate occurs in the cytosol via GSH which then enters lysosomes where disulfide interchange occurs, permitting the mixed disulfides to exit, and causing cystine depletion [5].

Laube et.al studied GSH in native cystinotic proximal tubular epithelial cells obtained from patient urine. They found an increased apoptotic rate of approximately 200% after TNF alpha stimulation as measured by TUNEL. Cystinotic proximal tubular cells had 6.8 nmol GSH/mg protein, whereas the controls had 11.8 nmol (*p* < 0.001) and impaired recovery from hypoxic stress [11]. These cells had a two-fold increase in apoptosis rate without an exogenous trigger, not seen in other studies in fibroblasts or RPTE cells [8,11,36]. GSH was measured specifically in Laube’s study via HPLC with electrochemical detection [37]. In extensive studies in cystinotic ciPTEC, tubuloids and zebrafish, a connection between ROS, apoptosis, and alpha ketogluterate was established. These characteristics of Ctns^−/−^ tissues were ameliorated by co-treatment with cysteamine and bicalutamide. This suggests a possible clinical use in patients which might improve the renal tubule defect [38].

## 6. Thyroid Effects of Apoptosis

Along with the more prominent effects of cystinosin deficiency in the kidney, untreated cystinosis is associated with hypothyroidism during childhood [39,40]. Studies using thyrocytes from *CTNS* knockout mice, determined that the basis of the hypothyroidism seen in cystinosis is multifactorial and that apoptosis plays a large role in this aspect of the phenotype. In addition to defective thyroglobulin synthesis, which is driven by the endoplasmic reticulum stress response, and unfolded protein response, there is progressive colloid exhaustion and impaired thyroglobulin proteolytic processing within lysosomes. Further, *CTNS*-deficient thyrocytes demonstrated a 4.5 fold increase in cell proliferation and an even more markedly increased rate of apoptosis [41]. This increased apoptosis is also likely triggered by the unfolded protein response limiting synthesis and secretion of thyroglobulin. Together these effects within thyrocytes provide an explanation for the hypothyroidism seen in individuals with cystinosis.

## 7. Pancreatic Effects of Apoptosis

Abnormal insulin secretion represents one of the later manifestations of cystinosis. Adults with cystinosis have been noted to have higher rates of diabetes post-transplant compared to adults who have undergone renal transplantation for other causes even when factors like corticosteroid use have been considered [42]. The role of apoptosis in pancreatic dysfunction was studied using *CTNS* knockdown within a beta-cell line. When *CTNS* was knocked down, the cells were noted to have cystine accumulation and altered redox potential suggesting that this model system appropriately captured the phenotype within this organ. Reactive oxygen species were hypothesized to have multiple effects including increased NFκB expression and within the mitochondria, there was reduced mitochondrial membrane potential and decreased ATP generation. Both increased NFκB expression and decreased mitochondrial membrane potential led to increased rates of apoptosis. In turn, increased apoptosis, in addition to decreased ATP generation led to decreased rates of insulin release [43].

## 8. Summary and Future Directions

As described in this review, the role of apoptosis in the pathogenesis of cystinosis is established. Other abnormalities that have been noted in cystinosis cell physiology include altered gene expression, increased autophagy, increased oxidative stress, inappropriate mitochondrial stress response, and abnormal lysosomal function. As shown in Table 1, this review has outlined how these mechanisms affect programmed cell death within specific experimental systems and what the effects cysteamine had on apoptosis.

What remains to be determined is the role of these other non-apoptotic paths, and if they also contribute to cell death. Unanswered questions include: What is primary in causing increased gene expression in cystinotic cells? Is lysosomal cystine storage essential for causing increased apoptosis or does the increased intralysosomal cystine content simply represent an epiphenomenon of the disease? If the increased lysosomal cystine content is not required for pathogenesis, why does long-term cysteamine depletion of cystine in patients yield an average doubling of native kidney survival? If elevated lysosomal molarity isn’t the cause of increased lysosomal size, what is? What is the primary trigger in the apoptotic cascade? If GSH concentration is normal in cystinotic tissue, how is a redox imbalance initiated and maintained? What accounts for the milder phenotypes in intermediate and ocular cystinosis, and why do they show a lower elevation in apoptosis for a given cystine content than the nephropathic lines? Does increased apoptosis cause increased autophagy or does increased autophagy cause increased apoptosis [44]? Which cells/tissues/organ systems/organisms, such as the Ctns^−/−^ mouse [45], best model the pathological abnormalities in the patient? What moderates the increased apoptosis so that generalized dysmorphology does not occur? New molecular biology techniques and imaging technology are promising methods to address these questions and possibly shed new light on additional treatment strategies.

## Figures and Tables

**Table 1 cells-11-00670-t001:** Summary of Cell Death in Cystinosis.

Model System	Mediator	Target	Effect on Apoptosis or Necrosis	Effect of Cysteamine	References (Year)
Human cystinotic fibroblasts	Native lysosomal cystine	PKC δ	Increased apoptosis 3 fold	Lowered cystine and normalized apoptosis	[7,9] (2002, 2006)
Human normal fibroblasts	CDME-induced increased lysosomal cystine	PKC δ	-	N/A	[7,9] (2002, 2006)
Human normal RPTC	CDME	PKC δ	Increased apoptosis 8 fold	N/A	[7,9] (2002, 2006)
Normalrat fibroblasts	CDME	N/A	Increase apopotosis 16 fold	N/A	[10] (2005)
Normal mouse fibroblasts	CDME	N/A	Increase apoptosis 5 fold	N/A	[10] (2005)
Cystinosis patient-derived induced pluripotent stem cells	Native lysosomal cystine	N/A	Increased apoptosis 1.4 fold	No effect	[12] (2020)
Ctns^−/−^ Zebrafish	Native lysosomal cystine	N/A	Increased apoptosis 7 fold	Decreased apoptosis	[6] (2017)
Cystinotic RPTC	Native lysosomal cystine	N/A	Increased apoptosis 2 fold	N/A	[11] (2006)
Cystinotic RPTC	Native lysosomal cystine	Caspase 4	Increased apoptosis 3 fold	N/A	[19] (2010)
Cystinotic RPTC	NLRP2	NF-κB	Increased apoptosis	N/A	[22] (2019)
Ctns(^−/−^) mice	N/A	Atubular glomeruli	Increased necrosis, apoptosis, and autophagy	N/A	[15] (2015)
siRNA knockdown of CTNS in normal RPTC	ROS	GSH, Redox capacity	Increased early and late apoptosis, and necrosis	Decreased apoptosis and necrosis	[33] (2016)
Cystinotic ciRPTC Ctns^−/−^ Zebra-fish larva	ROS	p62/SQSTM1	Increased apoptosis 3.5 fold Increased apoptosis 5 fold	Cysteamine not done Reversed by luteolin	[20] (2020)
Cystinotic ciRPTC Ctns^−/−^ mice Zebrafish	ROS	N/A	Increased apoptosis 7 fold	Reversed by cysteamine and disulfiram	[36] (2021)

CDME: Cystine dimethyl ester; RPTC: Renal proximal tubular cells; ciRPTC: Conditionally immortalized proximal tubule epithelial cells; ROS: Reactive oxygen species.

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
