# Peer review of "Programmed Cell Death in Cystinosis"

_cells, 2022, doi:10.3390/cells11040670_

Round 1
Reviewer 1 Report
The present review evaluated the role of apoptosis in cystinosis.
-The Abstract is not informative. No infomration is provided regarding the role of programmed cell death in cystinosis.
-The search algorithm should be described.
-No Figures are provided. The cellular pathways involved should be schematically depicted in order to be comprehensible.
-The results of previous research should be presented in tables.
-No strong conclusions are provided. Therefore, the contribution of this review in clinical or research practice is limited.
-The conclusions section consists mainly of questions rather than answers.
Reviewer 2 Report
- This review is acceptable. The authors provide a review for the topic of regulation of apoptosis in a rare lethal autosomal recessive disease- cystinosis. It should be informative for readers of this field, but the apoptosis is just one type of programmed cell death but apoptosis is not equal to programmed cell death. The definition seems not clear enough.
- There is no any table or figure in this review which made it inaccessible for reader to easier realize the apoptotic regulation of this disease.
- A few typographic mistakes should be revised before submission.
Reviewer 3 Report
The authors describe very clearly what is known so far about apoptosis in cystinosis and stand relevant questions in the field.
The most important pathways are described.
Minor comments: I understand that the authors have narrowed the literature to apoptosis in cystinosis, but in general, there is no consensus of increased autophagy in cystinosis, therefore relating increased apoptosis to increased autophagy or vice-versa could be misleading.
- As in most manuscripts, it would be nice to see a figure summarizing the work.
Round 2
Reviewer 1 Report
The authors failed to adequately revise their manuscript; therefore, I have to reject it.
Author Response
No further comments from reviewers.